# Diversity and Antibiotic Resistance of Triticale Seed-Borne Bacteria on the Tibetan Plateau

**DOI:** 10.3390/microorganisms12040650

**Published:** 2024-03-25

**Authors:** Xuan Hou, Jie Yang, Jinjing Xie, Shaowei Zhu, Zhenfen Zhang

**Affiliations:** Key Laboratory of Grassland Ecosystem, Ministry of Education, Pratacultural College, Gansu Agricultural University, Lanzhou 730070, China; houx@st.gsau.edu.cn (X.H.); yangjie@st.gsau.edu.cn (J.Y.); xiejj@st.gsau.edu.cn (J.X.); zhusw@st.gsau.edu.cn (S.Z.)

**Keywords:** triticale, seedborne bacterial diversity, antibiotic resistance, biofilm formation ability, motility, Tibetan Plateau

## Abstract

The Tibetan Plateau is located in southwestern China. It has many important ecological functions, such as biodiversity protection, and is an important grassland agroecosystem in China. With the development of modern agriculture and animal husbandry, antibiotics are widely used to treat humans and livestock, and antibiotics cannot be fully metabolised by both. Antibiotics eventually find their way into the environment, affecting other parts of grassland agroecosystems. Triticale (*Triticosecale wittmack*) is an artificial hybrid forage that can be used for both grain and forage. This study revealed the diversity of seedborne bacteria in triticale on the Tibetan Plateau and the resistance of the bacteria to nine antibiotics. It identified 37 representative strains and successfully obtained the spliced sequences of 36 strains of the bacteria, which were clustered into 5 phyla and 16 genera. Among them, 18 strains showed resistance to at least one of the 9 antibiotics, and the colony-forming unit (CFU) abundance of antibiotic-resistant bacteria (ARB) accounted for 45.38% of the total samples. Finally, the bacterial motility and biofilm formation ability were measured, and their correlation with bacterial resistance was analysed. The results showed that the bacterial resistance did not have an absolute positive correlation with the motility or biofilm formation ability.

## 1. Introduction

World agricultural land is decreasing, but the population is still increasing [1]. As a result, the world will need to produce more food on less land than before to meet people’s food needs. In order to prevent pests and diseases from reducing agricultural and animal husbandry production, antibiotics have to be used in large quantities in agricultural and animal husbandry production [2]. Antibiotics are overused, and the incidence of antibiotic-resistant bacteria (ARB) has increased sharply. The resulting antibiotic contamination has become a focus of concern [3,4,5,6]. The overuse of antibiotics in agriculture and animal husbandry is considered a major cause of the antibiotic resistance crisis [7]. In grassland agroecosystems, the microbiome drug resistance in the environment and animals has received widespread attention [8,9,10], but there are few reports on drug resistance in plant microbiomes. Seeds exhibit microbial diversity [11,12,13]. This is because the microorganisms are accompanied by vertical plant propagation and are planted in the seeds, and the microorganisms in the environment are planted on the seed surface. As a special organ, seeds have a wider drift range and a greater drift amount than plants. Seeds can not only continue the species but can also attach to the seeds and survive in the next generation of plants, which will affect the surrounding environment. ARB is also passed on to the seeds. However, the circulation range and volume of seeds in this system cannot be ignored, the ecological niche of the forage in the grassland agroecosystem cannot be ignored, and the flow of ARB throughout the system and between systems cannot be ignored.

Triticale occupies an important position in the grassland agricultural ecosystem of the Tibetan Plateau. Its seeds and plants can be used for livestock feeding, and it is an important link connecting the inorganic environment and livestock. A study by Dai [14] showed that the output, nutritional quality, feeding quality, and silage quality of small rye are greater than those of ryegrass (*Lolium perenne*) and oat (*Avena sativa* L.) on the Tibetan Plateau. However, the current main research on triticale still focuses on stress resistance, yield and feeding quality. However, studies on its role in bacteria, especially in terms of bacterial resistance, are rare. Understanding the intricate relationships among the environment, humans, animals and plants requires additional research to determine the impact of human activities on environmental resistance and the impact of ARB in the environment on human health. This study used triticale in the Tibetan Plateau as the research object and seeds as the material to determine the bacterial diversity, drug resistance, motility and biofilm formation ability of the cultivable seedborne bacteria of triticale, in order to understand the community composition and differences in seedborne bacteria among different varieties of triticale. Moreover, we study the correlation between drug resistance and the motility and biofilm formation ability of triticale-bearing bacteria. The number and proportion of ARB present in culturable triticale seedborne bacteria at the genus level were annotated to determine their impact on animals, the environment and humans.

## 2. Materials and Methods

### 2.1. Triticale Seeds

A total of eight varieties of triticale seeds were used in this study. Among them, those with a storage life of less than three years include Gannong No. 1, Gannong No. 2 and Zangsi No. 1, and those with a storage life of more than four years include Zhongsi 1048, Shida No. 1, Gannong No. 3, Gannong No. 4 and Gannong No. 7. All the seeds were produced on the Tibetan Plateau and provided by the Pratacultural College, Gansu Agricultural University. In order to improve readability, in this article, G1, G2, G3, G4, G7, Z2, S1 and ZS, respectively, represent the non-sterilised treatment of Gannong No. 1, Gannong No. 2, Gannong No. 3, Gannong No. 4, Gannong No. 7, Zangsi No. 1, Shida No. 1 and Zhongsi 1048; XG1, XG2, XG3, XG4, XG7, XZ2 and XZS, respectively, represent its disinfected treatment group.

### 2.2. Antibiotics and Media

Nine antibiotics were used in the present study. Tetracycline (active ingredient 95.0%, dissolvent: ddH_2_O, TET), ampicillin (USP grade, d: 1 mol/L HCl, AMP), ciprofloxacin (a.i. 90.0%, d: 1 mol/L NaOH, CPFX), sulfadiazine (a.i. 98.0%, d: DMSO, SUD), amikacin (a.i. 67.4%, d: ddH_2_O, AMI), oxytetracycline (a.i. 95%, d: 0.1 mol/L HCl, OTC), kanamycin (a.i. 75.0%, d: ddH_2_O, KAN) and erythromycin (a.i. 94.1%, d: ddH_2_O, EM) were kindly provided by Beijing Solarbio Science & Technology Co., Ltd., Beijing, China; rifampicin (a.i. 98.0%, d: DMSO, RFD) was generously provided by Wuhan Pytbio Bioengineering Co., Ltd., Wuhan, China. The above antibiotics were dissolved and diluted with the corresponding solvents to 50 mg/mL, a stock solution was prepared, and the mixture was stored at 4 °C for subsequent dilution in this study. Additionally, tryptone soybean agar medium (TSA) was used for the isolation and purification of bacteria; nutrient broth agar medium (NA) was used for the propagation of bacteria; 15% glycerol–nutrient broth medium (15% glycerol–NB) was used for the cryopreservation of bacteria; 0.3% agar–NB was used for the determination of bacterial motility; and Luria–Bertani medium (LB) and Luria–Bertani agar medium (LA) were used for the determination of bacterial resistance to antibiotics.

### 2.3. Separation, Purification and Preservation of Bacteria from Triticale Seeds

On the basis of Zhang’s method, the triticale seedborne bacteria were separated and divided into a surface disinfection treatment group and a non-disinfection treatment group [15]. Eight varieties of triticale seeds (1 g) were collected. In the surface disinfection treatment group, the seeds were placed in a sterile cup, 75% ethanol solution was added for soaking for 1 min, and the plants were subsequently rinsed 3 times with sterile water. We added 1% NaClO(aq) to soak for 2 min, and slowly shook the cup during the steeping process. Finally, sterile water was used to rinse the samples 3~5 times to remove the residual sterilisation solution. The last flushing fluid was coated on the culture medium, and the cells were cultured as the control group. The sterilised seeds were placed in a sterile mortar, 10 mL of sterile water was added, and the mixture was ground to a uniform pulp. After removing the mixture for 20 min, 1 mL of clear liquid was added to 10^−1^, 10^−2^, 10^−3^, 10^−4^, and 10^−5^ dilution gradient solutions. One hundred microliters of the original solution and the diluted gradient solution were added to a TSA medium coating plate, and the mixture was cultured for 24~48 h in the dark. Each process was repeated three times. The plants in the surface-free treatment group were directly placed in a sterile mortar, and sterile water was used as the control group.

The community diversity of the culture plates was determined.

All of the separations were picked in Petri dishes, and the three-line coating method was used for cultivation for 24~48 h in the dark. More than 3 generations of culture were cultivated continuously. After the shape of the strain was stable, representative separations were selected and saved.

Two millilitres of 15% glycerin-NB added to the frozen storage tube, after which the single bacteria were stably separated and stored at −80 °C.

### 2.4. Extraction and Appraisal of 16S RRNA from Triticale Seedborne Bacteria

This study used a Tiangen Bacterial DNA Extraction Kit (Tianten Biochemical Technology (Beijing) Co., Ltd., Beijing, China) to extract DNA from the triticale seedborne bacteria. The specific method used was described previously.

### 2.5. Determination of Bacterium Isolate Motility

The needle-punching process [16] was used to inoculate the strains in 0.3% agar–NB. After the plants were cultured in the dark for 20 h, the diameter of the turbid zone (mm) was measured, and the process was repeated three times.

### 2.6. Determination of the Biofilm Formation Ability of Bacteria

In this study, crystal violet staining [17] was used to determine the bacterial biofilm formation ability. Based on the OD_C_ control, the bacterial biofilm formation ability was divided into four levels: strong biofilm formation ability, OD_570_ > 4OD_C_; medium biofilm formation ability, 4OD_C_ ≥ OD_570_ > 2OD_C_; weak biofilm formation ability, 2OD_C_ ≥ OD_570_ > OD_C_; and no biofilm formation ability, OD_570_ ≤ OD_C_.

### 2.7. Determination of Antibiotic Resistance of Bacteria

According to the instructions for the use of antibiotics, we prepare them into stock solutions of corresponding concentrations. The stock solution concentration of ciprofloxacin and rifampicin is 25 mg/mL, and the stock solution concentration of other antibiotics is 50 mg/mL. A 2-fold dilution was used to set the antibiotic concentration gradient to 40, 80, 160, 320, and 640 μg/mL.

The qualitative screening of ARB LB antibiotic tablets (40 μg/mL) was prepared in order to paint the activated strain on the tablet, which was subsequently cultivated in the dark for 20 h, after which the growth of the strain was assessed. The growth is “+”; otherwise, it will be “−”. The strain that was recorded as “+” was used for future experiments.

The inhibitory zone method recommended by GB/T 38483-2020 [18] was used to determine the maximal inhibitory concentration (IC_50_) of antibiotics against triticale seedborne bacteria. To reduce the antibiotic use, the Oxford cups were replaced with paper pieces of the same diameter. The paper plugs with a diameter of 8 mm were soaked in prepared antibiotic solutions of 40, 80, 160, 320, and 640 μg/mL, and the control group was treated with the corresponding solvent. The activated strain was picked into LB and shaken at 37 °C overnight. The bacterial solution was diluted with LB until the OD_600_ was between 0.05 and 0.10. The diluted bacterial solution was mixed with LB agar at 55 °C, at a ratio of 1:9 to prepare a plate. After the plate had cooled and solidified, the prepared antibiotic paper was stuck onto the bacterial plate. After incubating in the dark at 37 °C for 20 h, the diameter of the inhibition zone was measured. This process was repeated 5 times.

The inhibition rate was calculated as follows:$$I=\frac{D_{1}-D_{2}}{D_{2}}\times 100\%$$

In the formula: *I*: Inhibition rate; *D*_1_: The diameter of the zone inhibited by the test antibiotic in the plate (units: mm); *D*_2_: The diameter of the inhibition zone formed by the blank control in the plate (units: mm).

Then, the drug-resistant strains were screened quantitatively. The inhibitory effects of each antibiotic at concentrations of 40 and 640 μg/mL were calculated. Strains with an IC_50_ < 40 μg/mL were considered “low-ARB”, and strains with an IC_50_ > 640 μg/mL were considered “super-ARB” and were not subjected to subsequent analysis. The bacteria with 40 < IC_50_ < 640 μg/mL were considered “ARB” and were subjected to a subsequent quantitative analysis of their IC_50_.

Finally, the IC_50_ of the ARB was quantitatively analysed. A regression analysis was performed based on the logarithm of the antibiotic concentration and the corresponding inhibition rate to obtain the regression curve Y = aX + b, and the conceptual value of the IC_50_ was calculated.

## 3. Results

### 3.1. The Characteristics of Triticale Seedborne Bacteria

#### 3.1.1. Isolation and Identification

According to the strain morphology, 37 representative strains with different colony morphologies were isolated from 8 triticale varieties.

After the DNA extraction from 37 isolates and PCR amplification, the amplified samples were sent to Shanghai Paison Biotechnology Co., Ltd., Shanghai, China, for sequencing, and 36 spliced sequences were obtained. The spliced sequences were compared via BLAST in the NCBI database “https://www.ncbi.nlm.nih.gov/” (accessed on 7 October 2023), and MEGA 11.0 and Adobe Illustrator 2022 were used to construct a phylogenetic tree (Figure 1), to determine the taxonomic status of the isolates.

Through a phylogenetic tree comparison analysis, 36 seedborne bacteria from 8 triticale varieties were clustered into 5 phyla and 16 genera.

The strains were Firmicutes: *Bacillus*: G1-7, XG1-3, G7-5, G7-4, XG1-8, XG7-6, G3-1, XG3-10, G4-11, XG4-8 and G3-4; *Exiguobacterium*: Z2-19; *Paenibacillus*: XZ2-1 and XG7-3; *Peribacillus*: S1-7 and Z2-5; *Psychrobacillus*: G1-1; and *Staphylococcus*: G2-10, G3-8, G4-9, G7-8, and XG3-8.

Actinobacteria: *Curtobacterium*: G7-7, ZS-2 and S1-4; *Microbacterium*: G1-5, G2-4 and G1-4; *Plantibacter*: G4-8.

Proteobacteria: *Erwinia*: Z2-6; *Pantoea*: G1-3; *Mixta*: ZS-1; *Acinetobacter*: G3-3; *Pseudomonas*: G2-17.

Bacteroidetes: *Chryseobacterium*: G2-16.

Unknown phyla. *Desemzia*: ZS-5.

#### 3.1.2. Bacterial Community Abundance Composition Analysis

After culturing in the dark for 48 h, a bacterial community analysis was performed on a total of 16 samples of 8 varieties of triticale. Based on the statistical results, the abundance composition and difference analysis of the 16 sample communities were conducted. Since the number of colonies on all gradient diversity plates of samples S1 and XS1 did not exceed 10 and did not meet the statistical standards of GB 20287-2006 [19], a subsequent analysis was not performed.

Since the differences in the bacterial diversity and abundance between samples S1 and XS1 were not statistically significant, only the diversity of the other 7 species and 14 samples were analysed. The results are shown in Figure 2. The bacterial abundances of samples G1, G2, and Z2 were relatively greater than those of the other samples, which were 1.97 × 10^7^ CFU/g, 1.65 × 10^7^ CFU/g and 1.37 × 10^7^ CFU/g, respectively. The bacterial abundance in the surface-undisinfected group was relatively greater than that in the disinfected group. Moreover, at the genus level, the abundance of *Bacillus* was relatively greater than that of the other genera (Figure 2A). *Bacillus* had the highest contribution to the composition of the total sample, followed by *Staphylococcus* and *Microbacterium*. The species abundance that may exist in the sample was estimated through the species accumulation curve (Figure 2B). The species abundance estimates of samples G1, Z2 and G2 are relatively greater than those of the other samples. Among them, samples G1, Z2, G2, G4 and XZ2 may contain more than 20 genera of culturable bacteria (Figure 2C). However, after an unsterilised and disinfected seed surface treatment, the bacterial species abundances of the seven varieties of triticale (Figure 2D) are at the genus level, with up to 8 genera isolated for each variety. There was one common genus, *Bacillus*. LDA was performed on 14 samples (Figure 2E). At the genus level, the intergroup differences between sample G1 and the other samples were caused mainly by *Bacillus*; the intergroup differences between sample G3 and the other samples were caused mainly by *Staphylococcus* (*p* < 0.05).

#### 3.1.3. Differences in Culturable Triticale Seedborne Bacterial Communities

The results of the differential analysis of the culturable seedborne bacterial community composition of the seven varieties of triticale are shown in Figure 3. The principal component analysis (PCA) results are shown in Figure 3A. Among the community compositions among samples G3, G4, G7 and ZS, the differences in the abundance of important species are small, and the community compositions are common; however, the differences in the abundances of characteristic species in the community compositions of samples G1, G2, Z2 and other samples are relatively large. The difference between sample G2 and the other samples is mainly caused by PC2; the difference between samples G1 and Z2 and the other samples is mainly caused by PC2. The principal coordinate analysis (PCoA) results are shown in Figure 3B. The similarity in the community species composition between samples G1, Z2, G2 and G4 was greater than that between other samples; the similarity in community species composition between samples G3 and G7 was greater; and the community species composition in sample ZS was greater. The species composition was relatively different from that of the other samples (*p* < 0.05).

#### 3.1.4. Ecological Niche Analysis of Culturable Triticale Seedborne Bacteria

Compared with that in the unsterilised group, the bacterial abundance in the disinfected group of triticale seeds of the same variety decreased. Notably, the species abundance at the genus level did not completely decrease. Some genera did not appear on the TSA media in the unsterilised group but appeared after disinfection. After reviewing the information, a reasonable explanation given for this phenomenon was competition for ecological niches. In the non-disinfection treatment group, the abundance of all the species remained at a high original level, the abundance of the dominant bacterial genera was greater, and the competitive ability was greater than that of the nondominant bacterial genera; however, the nutrients and space in the TSA medium were limited. Therefore, non-dominant bacterial genera in lower ecological niches cannot compete for nutrients or space with dominant bacterial genera in high ecological niches at high abundance and therefore cannot grow in this medium. However, after a surface disinfection of the seeds, the abundance of all of the bacteria in the seeds decreased (including dominant bacterial genera and non-dominant bacterial genera). This treatment not only alleviated the competitive pressure at the genus level but also alleviated the internal competitive pressure. As a result, non-dominant bacterial genera in low ecological niches can compete for nutrients and space under the low abundance of dominant bacterial genera, causing them to appear on the TSA medium of the disinfection treatment group, and even causing the dominant bacterial genera to change.

Based on these findings, bacterial ecological niches were compared at the genus level by evaluating the upregulation and downregulation of bacteria in the unsterilised group of the same species in the disinfected treatment group. In G1 and G2, no species with an increased abundance appeared. In G3, there were more ecological niches for *Acinetobacter* and *Staphylococcus* than for *Pantoea*, *Pseudomonas* and *Plantibacter*; in G4, there were more ecological niches for Plantobacter and *Staphylococcus* than for *Curtobacterium* and *Desemzia*; in G7, the ecological niches for *Desemzia*, *Staphylococcus* and unknown bacteria were greater than those for *Bacillus*, *Exiguobacterium*, *Paenibacillus* and *Pantoea*; in Z2, the ecological niches for Desemzia, *Exiguobacterium*, *Microbacterium* and *Staphylococcus* were greater than those for *Pantoea*; and in ZS, the ecological niches for *Desemzia* were greater than those for *Bacillus*, *Curtobacterium*, *Exiguobacterium* and *Microbacterium* (Figure 4) (*p* < 0.05).

### 3.2. Biofilm Formation Ability

The biofilm formation ability coefficient (BFAC) of culturable triticale seedborne bacteria was calculated as the ratio of the OD_570_ value to the OD_C_. The study revealed that there are 18 strains with a strong biofilm formation ability; 7 strains with a medium biofilm formation ability; 11 strains with a weak biofilm formation ability; and only 1 strain with a poor biofilm formation ability, G4-8 (Figure 5). Among them, strain G7-7 has the strongest biofilm formation ability and belongs to the genus *Curtobacterium*; G4-8 has the weakest biofilm formation ability and belongs to the genus *Plantibacter*.

### 3.3. Motility

The results of the determination of bacterial motility in the triticale seed belts are shown in Figure 6. Among them, the strains with the highest level of motility were XG7-6 and G3-4, with a motility diameter of 90 mm; followed by strains G4-9 and G4-11. There were 14 strains with the lowest level of motility, namely, G1-1, S1-7, G2-16, G4-8, G7-8, G2-10, G3-3, G7-7, G1-5, S1-4, ZS-2, ZS-5, and G2-4, whose movement diameters were less than 5 mm. Among them, the strains XG7-6, G4-11 and G3-4, which have a high motility, all belong to the genus *Bacillus* (*p* < 0.05), and G4-9 belongs to the genus *Staphylococcus*; however, among the 13 strains with low motility, 12 strains were successfully clustered into the genus, and none of these 12 strains belonged to the genus *Bacillus* (Figure 6A).

The Euclidean distance cluster analysis was also conducted on the motility of 37 strains of bacteria. Based on a Euclidean distance of 1.8, the data were divided into 4 clusters (*p* < 0.05). From top to bottom, they are cluster I, cluster II, cluster III and cluster IV, with athletic abilities ranging from weak to strong. Among them, cluster I contained 16 isolates, G3-8, G7-5, etc.; cluster II contained 15 isolates, G1-3, G1-4, etc.; cluster III contained isolates G4-11, G4-9, XG4-8 and Cluster Z2-5; and IV contained isolates G3-4 and XG7-6. Its motion diameter ranged from weak to strong and ranged from 1.00 ± 0.06~14.67 ± 0.88 mm, 16.5 ± 0.29~31.17 ± 0.60 mm, 41.00 ± 0.58 mm~55.33 ± 1.45 mm, and 90 ± 0.06 mm (Figure 6B).

### 3.4. Bacterial Resistance to Antibiotics

#### 3.4.1. Qualitative Screening

The qualitative screening results are detailed in Appendix A. On LB agar supplemented with an antibiotic concentration of 40 μg/mL, a total of 22 strains were unable to grow in 9 antibiotic media. The strains with the most severe resistance at this concentration were G2-16 and XG3-8; these strains can grow normally on six antibiotic plates and are sensitive to only three other antibiotics. There are seven strains that can only grow on one antibiotic plate. The results showed that the triticale seedborne bacteria were resistant to low concentrations of multiple antibiotics.

Among the nine antibiotics tested, tetracycline, ciprofloxacin, and oxytetracycline at 40 μg/mL had broader antibacterial effects, and only one strain, G3-8, XG3-8 and G3-8, grew on the plates. Notably, strain G3-8 can grow on only tetracycline and oxytetracycline plates and is sensitive to seven other antibiotics at these concentrations. This phenomenon needs further research. Ampicillin had the worst antibacterial ability, with only 23 susceptible strains.

#### 3.4.2. Quantitative Screening

Since G2-16 belongs to *Chryseobacterium*, its drug resistance cannot be analysed by the inhibition zone method [20], so only the remaining 36 strains were quantitatively analysed.

The results of the quantitative screening calculations are detailed in Appendix A.

After a quantitative analysis and screening, the IC_50_ of 40~640 μg/mL were quantitatively determined. See Table 1 for details.

#### 3.4.3. Quantitative Analysis of the IC_50_

A regression equation was constructed for the strains listed in Table 1 at five antibiotic concentration gradients and for the corresponding antibacterial rates. The results are shown in Table 2. The R^2^ values of all the regression equations are greater than 0.8. According to the statistical analysis, the regression equation has a high degree of fit, and the IC_50_ value has a high credibility.

After calculation, the conceptual IC_50_ value of strain G2-17 against kanamycin was 365 μg/mL, the conceptual IC_50_ value of strain G3-8 against tetracycline was 399 μg/mL, and the conceptual IC_50_ value of strain XG3-8 against erythromycin and ciprofloxacin was 399 μg/mL. The conceptual IC_50_ values of the stars are 427 and 177 μg/mL, respectively.

#### 3.4.4. Visualisation of the Resistance Analysis Results

The visualisation of the drug resistance of the bacterial isolates from all the triticale species except strain G2-16 is shown in Figure 7. Strains G1-4 and G7-8 were clustered into one category and were extremely resistant to only erythromycin; strains ZS-5, XG1-3, S1-7, G2-10 and G4-9 were clustered into one category. The strains are highly resistant to ampicillin only; strains G4-11 and Z2-5 are clustered into one category, which are highly resistant to only sulfadiazine; strains ZS-1, S1-5 and XG4-8 are clustered into one category of strains, all of which are highly resistant to two antibiotics, ampicillin and sulfadiazine; and strains G4-7 and G2-17 are clustered by property. The strains G4-9, G2-10, S1-7, XG1-3 and ZS-5 were clustered into one category, all of which were highly resistant to ampicillin.

The effective concentration of antibiotics was 40 μg/mL. Rifampicin had inhibitory effects on all 36 strains. Tetracycline and oxytetracycline had inhibitory effects on 35 strains, except G3-8. Kanamycin had inhibitory effects on all the strains except G2-17. Ciprofloxacin has inhibitory effects on all bacteria except for XG3-8.

In summary, rifampin, tetracycline, oxytetracycline, kanamycin and ciprofloxacin have relatively broader spectra and more effective inhibitory effects than ampicillin, sulfadiazine, amikacin and erythromycin. Erythromycin, ampicillin and sulfadiazine are three different antibiotic categories: macrolides, β-lactams and sulfonamides, respectively. Therefore, strains G4-7, G2-17 and XG3-8 are MDARB.

The 37 strains isolated, excluding G2-16 of the genus *Chryseobacterium*, cannot use the inhibition zone method to estimate the conceptual value range of their IC_50_. Among them, 18 strains had strong effects on one or more of the 9 mainstream antibiotics, or super-drug-resistant strains, accounting for 50% of the strains tested, were clustered into *Desemzia*, *Curtobacterium*, *Pseudomonas*, *Staphylococcus*, *Microbacterium*, *Bacillus*, *Mixta* and an unknown genus. The ARB were mainly concentrated in *Staphylococcus* and *Bacillus*, with five and three strains, respectively. The five strains that were clustered into the *Staphylococcus* genus were all resistant. Among the drug-resistant strains, 3 strains were MDARB, accounting for 16.7%, namely, *Pseudomonas*, *Staphylococcus* and an unknown genus.

### 3.5. Correlations between Resistance and Motility and Biofilm Formation Ability

An intragroup correlation analysis was conducted based on the results of the resistance of 36 strains to 9 antibiotics and the results of motility and biofilm formation ability measurements. The biofilm formation ability of triticale culturable seedborne bacteria was significantly negatively correlated with a bacterial resistance to erythromycin, and bacterial motility was significantly positively correlated with a bacterial resistance to sulfadiazine. Moreover, the biofilm formation ability and motility have no significant impact on other bacterial resistances (Figure 8).

### 3.6. Analysis of Bacterial Resistance and Diversity

According to the statistics, the 14 triticale seed samples contained a total of approximately 6.88 × 10^7^ CFU, of which 3.21 × 10^7^ CFU of ARB accounted for 45.38% of the total sample abundance. The distribution of the abundances of ARB of different genera in each sample is shown in Figure 9. The abundance of ARB in samples G2, Z2 and G1 was relatively greater than that in the other samples. Among the 14 samples, the abundance of ARB in 7 samples reached more than 50%, and the abundance of drug-resistant bacteria in two samples was close to 50%. The strains that clustered into *Mixta*, *Desemzia*, *Exiguobacterium*, *Pseudomonas*, *Staphylococcus*, *Microbacterium* and unknown genera all had varying degrees of drug resistance; *Curtobacterium* and *Bacillus* are resistant to some strains.

## 4. Discussion

Plants are producers in the ecosystem, and plant seeds not only enable the continuation of the species but also allow microorganisms to spread vertically among the plants [21]. In this study, the 36 representative isolates were clustered into 5 phyla and 16 genera. Like the diversity of seedborne bacteria in plants such as barley (*Hordeum vulgare*) [22], passion fruit (*Passiflora edulis*) [23], wheat (*Triticum aestivum*) and corn (*Zea mays*) [24], the seedborne bacteria are mainly clustered into Firmicutes, Actinobacteria, Proteobacteria and Bacteroidetes. However, the dominant bacterial phyla vary depending on plant species, origin, etc. Studies have shown that some plant seedborne bacteria have antagonistic effects on plant pathogens and biocontrol functions [25,26]. These include *Bacillus* species, which are 2.8 times more potent than seed fungi and are commonly found in plant seeds [25]. *Bacillus* was the dominant bacterial genus in this study. Therefore, the use of seeds from the 37 preserved strains of triticale may have a biological control significance.

After disinfection treatment, compared with those in the non-disinfection treatment group, the bacterial abundances in all the samples were lower on TSA plates, but the species abundances did not show a complete downwards trend. This study used an ecological theory to explain this phenomenon, and it evaluated this phenomenon to reveal the ecological niche of bacteria in seeds at the genus level. The following hypothesis was proposed through this study: The environmental stress reduced bacterial abundance and interbacterial competitive pressure in this sample, as compared to the original sample. This resulted in less competitive bacterial genera that could not have been isolated in the original sample, but could be isolated in the environmentally stressed sample. With the same sample, this method can isolate more bacterial resources than traditional methods. The disadvantage of these methods is that they cannot objectively determine the abundance of less competitive bacteria in the original sample. In this study, the genera *Curtobacterium* and *Desemzia* were found in sample XG4, and *Desemzia*, *Mixta* and *Paenibacillus* were found in sample XG7. None of these genera were found in the corresponding non-sterilised treatments. Supplementing Gannong No. 4, Gannong No. 7 and other varieties of triticale through a disinfection treatment can cultivate seedborne bacterial species.

Relevant studies have shown that the drug resistance of biofilm bacteria is hundreds or even 1000 times greater than that of planktonic bacteria [27]. However, the results of this study show that there is no significant positive correlation between bacterial resistance and biofilm-forming ability. This shows that the ability of biofilms to enhance bacterial resistance does not mean that the ability to form biofilms has an effect on improving the resistance level of planktonic bacteria. In other words, in the process of the development of planktonic bacteria into biofilm bacteria, neither the biofilm nor biofilm formation ability are the main factors affecting bacterial resistance. At this stage, the factors regulating bacterial resistance remain to be studied.

Motility not only provides virulence factors for bacteria, but also helps bacteria seek advantages, avoid disadvantages, adsorb and other functions, thus providing motile non-drug-resistant *E. coli* with a drug resistance that is 50 times higher than the lethal concentration of immotile *E. coli*. The premise is that the bacterial population invading the antibiotic area needs to reach a threshold density [28]. In the present study, bacterial motility and resistance were generally not positively correlated. Perhaps motility is an influencing factor in the development stage of planktonic bacteria into biofilm bacteria. However, since this experiment was testing drug resistance, the bacterial concentrations were all consistent. This leads to the possibility that the invasion concentration of some bacteria has reached the density threshold, making motility play a certain role in bacterial resistance. There are also some bacteria whose invasion concentration does not reach the density threshold, so the current results are shown. The density threshold for different motile bacteria to invade different antibiotics remains to be studied. Cutugno [29] proposed classifying the motility as none (<5 mm), medium (5~20 mm), or strong (>20 mm) in 0.8% agar–MH media. However, this method is not suitable for this study, and for the method used in this study, there is no clear basis for classifying the strength of bacterial motility. Therefore, this study conducted a Euclidean distance cluster analysis based on the results of 37 strains measured using a pinprick process in 0.3% agar NB, and classified the bacterial motility into weak (D < 15 mm), medium (15 ≤ D < 40 mm) and strong (D ≥ 40 mm).

Among the statistically significant samples, this study measured the resistance of triticale culturable species on the Tibetan Plateau to nine mainstream antibiotics and revealed that the abundance of ARB in the total samples was 3.21 × 10^7^ CFU, accounting for 45.38%. These ARB were clustered into eight genera, and these data were obtained from only 14 g of triticale seeds. The ARB abundance was the highest in sample G2, reaching an astonishing 85.14%. Triticale is a forage that can be used both as grain and forage. The aboveground part of the plant is an important part of livestock rations, and its seeds are often used in flour processing and beer brewing. Therefore, triticale is often eaten directly or indirectly. On the Tibetan Plateau, the annual fresh (dry) forage yields of triticale are 34.84 t/hm^2^ and 12.96 t/hm^2^, respectively [14]. According to incomplete statistics from the Ministry of Agriculture and Rural Affairs of China (PPC) “www.agri.cn/sj/ (accessed on 7 February 2024)”, the triticale planting area on the Tibetan Plateau exceeds 35,000 hectares. These include corporate planting and the distribution of grass products to Shaanxi, Gansu, Ningxia, Inner Mongolia, Tibet and other places. Due to the complex functions of the digestive system, foodborne ARB may not necessarily colonise the next trophic level directly, but bacteria are very good at sharing resistance genes. Resistance genes can spread rapidly within the same bacterial genus, even if the ingested bacteria remain in the gut for only a short time [30]. However, ARB in triticale seeds were found to be concentrated mainly within these four genera: *Bacillus*, *Staphylococcus*, *Pseudomonas* and *Microbacterium*. The related research shows that these four genera can all colonise the environment [31] (soil and water), agricultural animals [32,33,34,35] (bovine, sheep and pig) and humans [36,37]. Therefore, these four genera have a high possibility of colonizing the environment, agricultural animals and humans through the food chain, and influencing the microbial resistance within them. This speculation requires further research and verification. ARB flow to the next trophic level along the food chain, which will eventually pose a serious threat to livestock production and human health. However, high trophic levels are not the final destination of ARB. ARB return to the environment through the metabolism of humans and livestock and eventually circulate in grassland agroecosystems. Although the distribution of seedborne bacteria in plants is still unclear, a large amount of pasture carrying ARB is provided to livestock for consumption, and the drug-resistant bacteria are metabolised out of the body through livestock. Through this cycle, drug-resistant bacteria continue to integrate ARG with each other, which also affects the bacterial resistance of pasture and livestock, making the treatment of bacterial diseases more difficult and costly. This will also radiate and affect the surrounding environment (including people) [38]. ARB will also flow into other grassland agroecosystems through trade in triticale forage products and seeds. The Tibetan Plateau is the “Water Tower of Asia”, with eight major rivers delivering an average of 117.92 billion m^3^ of water downstream. Relevant data also show that a variety of antibiotic residues and the antibiotic resistance gene (ARG) has been detected in water bodies in the Tibetan Plateau [39,40,41,42]. This will also have an impact on downstream grassland agroecosystems. Therefore, blocking the spread of ARB, and more specifically the spread of ARG, is of great significance to the global antibiotic crisis and human health.

In summary, ARB in humans and livestock are not produced entirely under the pressure of antibiotics. Foodborne ARB are also important for humans and agricultural animals to obtain ARB from the environment. The occurrence of ARB is irreversible. There is currently no effective method for removing ARG from a system. Combined with the trophic level hypothesis, the toxins at high trophic levels are often more abundant than those at low trophic levels through enrichment. All drug resistance genes will eventually be integrated into human microorganisms, which will cause a disaster. It is important to make choices that respect natural pressures and reduce the unnecessary use of antibiotics. This strategy will reduce the profitability of livestock farming on a short time scale. However, on a long-term time scale, reducing the unnecessary use of antibiotics has important strategic significance for agricultural and animal husbandry production and human health protection.

## 5. Conclusions

This study describes the composition of the seedborne bacterial communities of different triticale varieties and the differences between them. The niche of triticale seedborne bacteria on TSA media was revealed by the change in bacterial abundance before and after seed surface disinfection. The results showed that there was no significant positive correlation between bacterial resistance or motility and biofilm formation ability. ARB accounted for a greater proportion of bacteria among the triticale seedborne bacteria. This will have a serious adverse impact on the grassland agroecosystem.

## Figures and Tables

**Figure 1 microorganisms-12-00650-f001:**
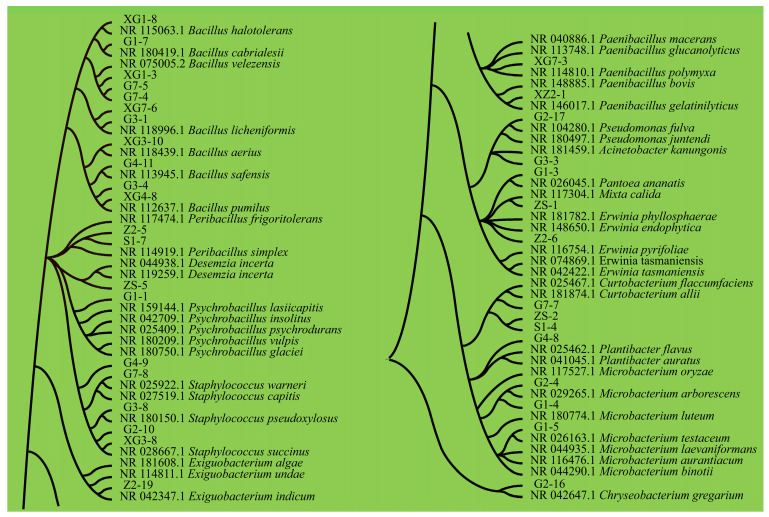
Phylogenetic evolutionary tree of the triticale seedborne bacteria.

**Figure 2 microorganisms-12-00650-f002:**
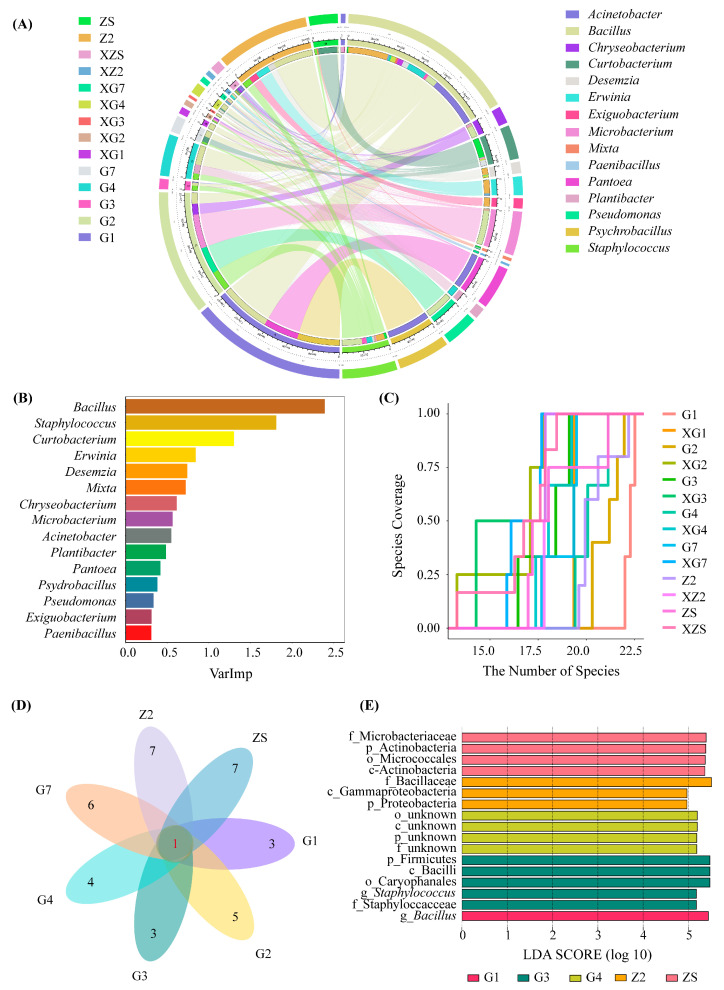
Analysis of composition and abundance in triticale culturable seedborne bacteria. (**A**) Species relationship of triticale culturable seedborne bacteria at the genus level; (**B**) random forests in the total sample at the genus level. (**C**) Species accumulation map of bacterial genus level in culturable seed-borne bacteria of triticale. (**D**) Venn of 7 varieties of triticale fruits with culturable seedborne bacteria at the genus level. The red numbers represent the number of bacterial genera shared by all triticale cultivars, and the black numbers represent the number of other bacterial genera isolated in each cultivar in addition to the bacterial genera shared. (**E**) LDA between samples. In the figure, G1 and XG1 represent the unsterilised treatment group and the disinfected treatment group, respectively, of the same variety (Gannong No. 1); other plants are similar to this one.

**Figure 3 microorganisms-12-00650-f003:**
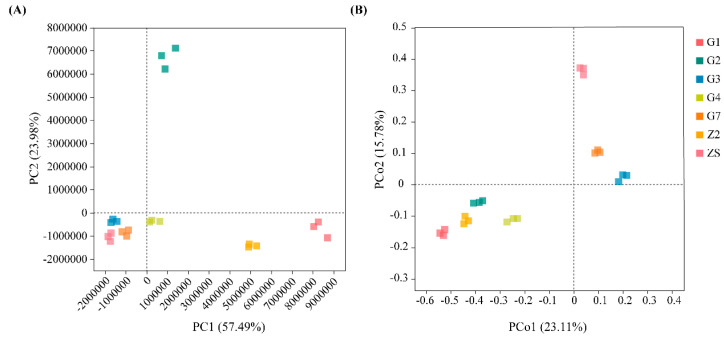
Analysis of differences in bacterial community composition among seven varieties of triticale. (**A**) PCA among triticale varieties; (**B**) PCoA among triticale varieties.

**Figure 4 microorganisms-12-00650-f004:**
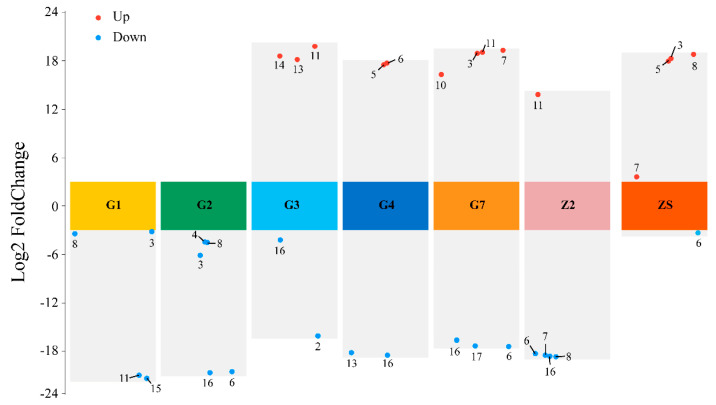
Volcano plot of the seedborne bacterial abundance of 7 varieties of triticale under disinfection treatment. The numbers in the figure represent different genera of bacteria. 1: *Erwinia*; 2: *Acinetobacter*; 3: *Bacillus*; 4: *Chryseobacterium*; 5: *Curtobacterium*; 6: *Desemzia*; 7: *Exiguobacterium*; 8: *Microbacterium*; 9: *Mixta*; 10: *Paenibacillus*; 11: *Pantoea*; 12: *Peribacillus*; 13: *Plantibacter*; 14: *Pseudomonas*; 15: *Psychrobacillus*; 16: *Staphylococcus*; 17: other.

**Figure 5 microorganisms-12-00650-f005:**
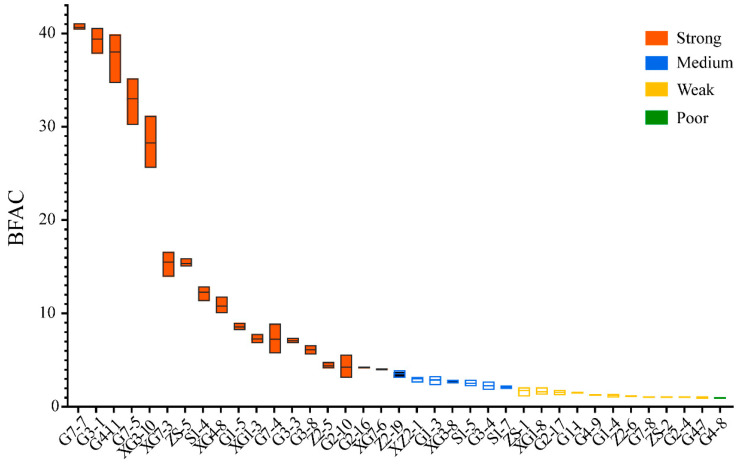
Biofilm formation ability of culturable triticale seedborne bacteria.

**Figure 6 microorganisms-12-00650-f006:**
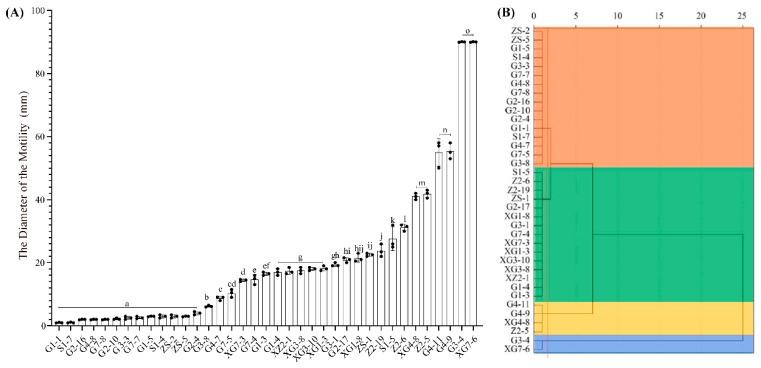
Motility cluster analysis of culturable triticale seedborne bacteria. (**A**): The histogram of bacterial movement diameter; (**B**): The cluster analysis of bacterial movement diameter. In (**A**), letters represent bacterial motility levels, letters from “a” to “o” represent bacterial motility levels from low to high, and the levels of adjacent letters are significantly different (*p* < 0.05); the orange part in (**B**) is cluster I, the green part is cluster II, the gold part is cluster III, and the blue part is cluster IV, and its bacteria mobility also changes from weak to strong.

**Figure 7 microorganisms-12-00650-f007:**
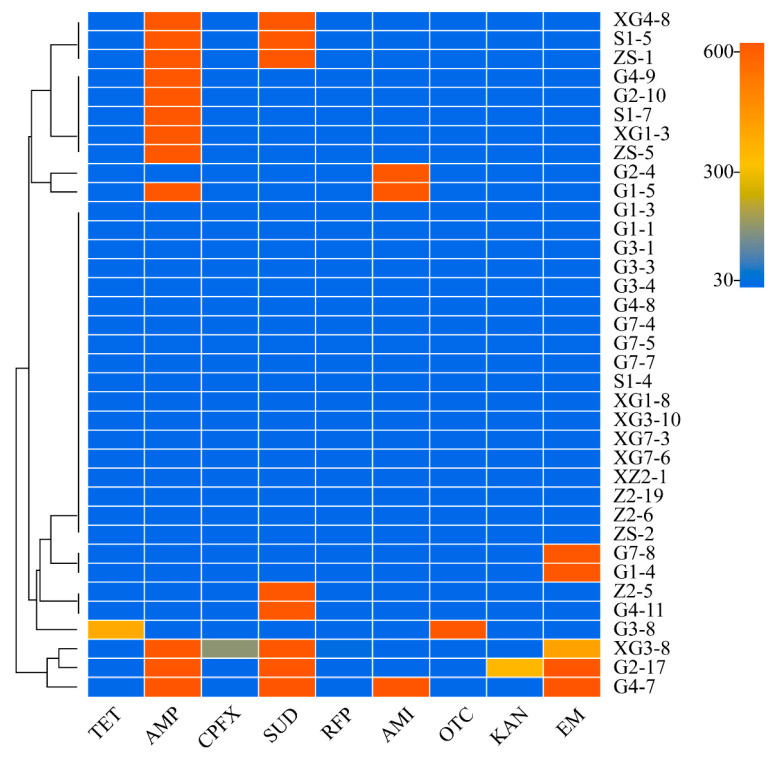
Heatmap of bacterial resistance to antibiotics.

**Figure 8 microorganisms-12-00650-f008:**
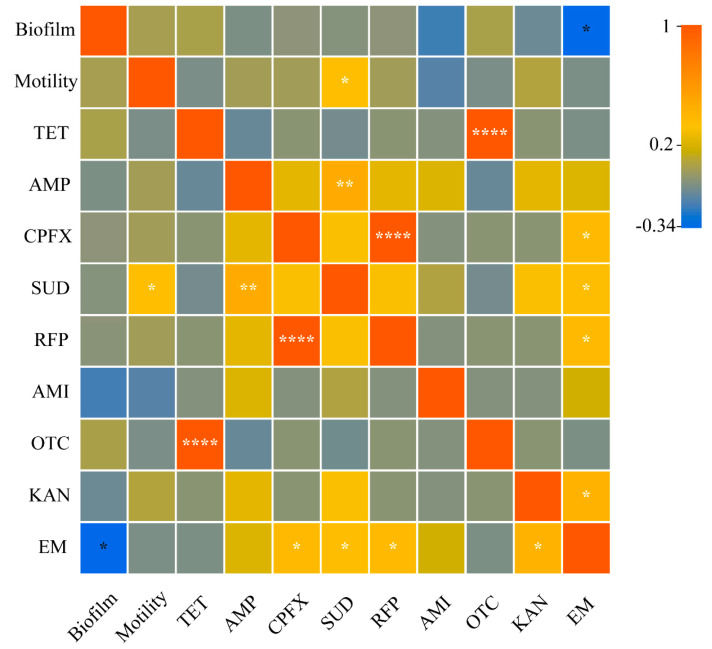
Correlations between triticale culturable seedborne bacterial biofilms and motility and between triticale and resistance to nine antibiotics Among them, *p* > 0.05; “*” indicates *p* < 0.05; “**” indicates *p* < 0.01; and “****” indicates *p* < 0.0001.

**Figure 9 microorganisms-12-00650-f009:**
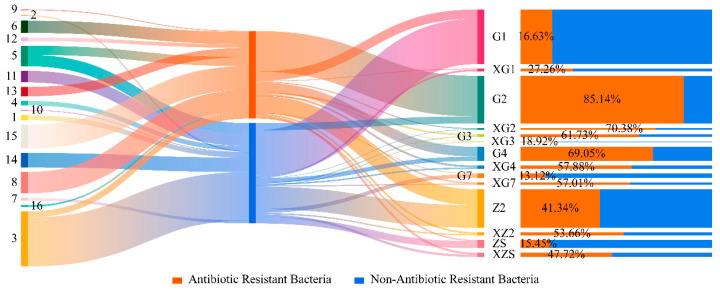
Distribution of ARB in the samples. The numbers in the picture on the left represent different genera of bacteria (1: *Erwinia*; 2: *Acinetobacter*; 3: *Bacillus*; 4: *Chryseobacterium*; 5: *Curtobacterium*; 6: *Desemzia*; 7: *Exiguobacterium*; 8: *Microbacterium*; 9: *Mixta*; 10: *Paenibacillus*; 11: *Pantoea*; 12: *Peribacillus*; 13: *Pseudomonas*; 14: *Psychrobacillus*; 15: *Staphylococcus*; 16: Unknown genus).

**Table 1 microorganisms-12-00650-t001:** Strain that fit the linear regression analysis.

Strain	Antibiotics				Inhibition Rate (*I*)	
	Name	Antibiotic Concentration (μg/mL)			40	640
		CK	40	640		
G2-17	KAN	8.00	9.38 ± 0.166	12.60 ± 0.100	17.3%	57.5%
G3-8	TET	8.00	9.70 ± 0.200	13.10 ± 0.400	21.3%	63.8%
XG3-8	CPFX	10.30 ± 0.224	10.92 ± 0.287	18.72 ± 0.185	6.0%	81.7%
XG3-8	EM	8.00	8.50	13.10 ± 0.400	6.3%	63.8%

**Table 2 microorganisms-12-00650-t002:** IC_50_ of different isolates with corresponding antibiotics.

Strain	Antibiotics						Linear Regression Analysis		
	Name	Bacteriostatic Rate Corresponding to Antibiotic Concentration					Regression Equation	R^2^	IC_50_
		40	80	160	320	640			
G2-17	KAN	17.30%	31.50%	38.80%	47.50%	57.50%	Y = 32.023X − 32.063	0.9881	365
G3-8	TET	21.30%	21.30%	33.80%	41.30%	63.80%	Y = 32.973X − 35.757	0.8703	399
XG3-8	EM	6.30%	15.00%	27.00%	39.30%	63.80%	Y = 46.274X − 71.714	0.9586	427
XG3-8	CPFX	6.02%	24.47%	53.59%	69.90%	81.75%	Y = 65.405X − 97.015	0.9784	177

## Data Availability

Data are contained within the article.

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
