# Peer review of "Diversity and Antibiotic Resistance of Triticale Seed-Borne Bacteria on the Tibetan Plateau"

_microorganisms, 2024, doi:10.3390/microorganisms12040650_

Round 1

Reviewer 1 Report

Comments and Suggestions for Authors

The work by Hou titled "Diversity and antibiotic resistance of triticale seed-borne bacteria on the Tibetan Plateau" involves the isolation of hundreds of endophytic bacterial strains from seeds and the selection of 37 of them for a deeper characterization of their diversity, biofilm production, motility, and antibiotic resistance.

Overall, the work is well-written, and there don't seem to be any issues with the English, although I'm not a native speaker.

The results are generally well-described, and the Discussion partially. Below are some suggestions:

1. The authors mention that they isolated over a thousand colonies from Triticosecale wittmack seeds, and their subsequent selection method was quite arbitrary, based solely on morphology. While this may be valid, I suggest that if there is no other scientific basis, they should only mention that 37 isolates were selected without describing all colonies.

2. One of the (negative) results found by the authors was that there is no correlation between biofilm formation or motility and antibiotic resistance. This contradicts several articles that have supported these correlations. How can this be explained? I suggest improving the Discussion and mentioning several reasons why such correlations were not found. I can suggest that these endophytes are not really pathogens; in fact, it's a characteristic of them not to cause apparent harm to their hosts. Perhaps they could suggest future experiments to evaluate this aspect.

3. I suggest extensively modifying and citing articles on biofilm formation and antibiotic resistance in the Discussion paragraphs 443-455. And please, avoid repeating results in the Discussion.

4. The paragraphs in the Discussion from 489 to 495 also require citations to substantiate the arguments they mention.

Other minor comments:

5. The phylogenies graph needs improvement; it looks blurry, and the legends cannot be perceived.

6. Correct the scientific name in the abstract: Triticosecale Wittmack. it is Triticosecale wittmack.

7. Please rewrite the phrase: "There was no significant correlation between their ability and biofilm formation ability and motility." Line 383-384. It is confusing.

Reviewer 2 Report

Comments and Suggestions for Authors

Abstract

The transition from the general discussion about antibiotics to the specific focus on triticale (line 13-14) is somewhat abrupt. It could be improved by providing a sentence that connects the two topics, explaining why the study focuses on triticale in the context of antibiotic use and resistance.

In line 15, "revealed the diversity of triticale seedborne bacteria and the resistance to 9 antibiotics" might be clearer if rephrased to specify that the study investigates both the diversity of bacteria and their resistance to antibiotics, rather than the resistance of the diversity itself.

The sentence in lines 15-16 could be more concise and clear. Consider rephrasing to something like: "This study analyzed the diversity of bacteria in triticale seeds and their resistance to 9 antibiotics, identifying 37 representative strains."

In line 16, "36 isolates were successfully clustered into 5 phyla and 16 genera" could be confusing because it previously mentioned 37 representative strains. It's unclear why there's a sudden shift to 36 isolates. Clarification or correction is needed here.

The phrase "18 strains were resistant to one or more of the 9 antibiotics" (line 17) could be simplified to "18 strains showed resistance to at least one of the 9 antibiotics" for better readability.

The abbreviation "CFU" (colony-forming units) in line 18 is used without being defined. 

Introduction

The introduction's transition from discussing antibiotic resistance to the challenges of global agriculture due to shrinking land and population growth needs better integration to maintain focus on the central theme of antibiotic resistance in agroecosystems.

The text redundantly emphasizes the overuse of antibiotics in agriculture and its contribution to resistance, which could be streamlined for conciseness.

The mention of past research on soil and fecal microbiomes' drug resistance lacks a direct link to the current study's focus on forage seed bacteria, leaving the relevance of these studies unclear.

The introduction of triticale and its attributes is swift, without a clear explanation of its relevance to the study's investigation into bacterial resistance, making the transition feel disjointed.

The objectives and rationale for studying triticale's seed-borne bacteria and their resistance traits are not clearly explained.

Materials and Methods

The naming convention for triticale seed varieties switches from specific names like "Gannong No. 1" to codes like "G1", "G2", etc., without a clear explanation or consistency throughout the text.

When discussing the preparation of antibiotics, the text mentions diluting them to a stock solution of 50 mg/mL without specifying whether this concentration applies to all antibiotics, given their varied solubilities and effective concentrations.

In the section on separating and purifying bacteria from triticale seeds, the description of the surface disinfection process is slightly confusing, particularly the phrase "slowly shake the toasting during the soaking process". It's unclear what "toasting" refers to, and this could be a typographical error or an unclear term.

The method for testing antibiotic resistance seems repetitive, mentioning the use of a qualitative screening followed by the determination of the maximal inhibitory concentration (IC50) using a method that seems overly complicated with the replacement of Oxford cups with paper pieces.

Results

The section discussing bacterial community abundance composition analysis (lines 192-216) mentions that samples S1 and XS1 were excluded from subsequent analysis without providing a clear rationale for their exclusion or detailing the "statistical requirements" they failed to meet.

In the biofilm formation section, the text redundantly lists strains with different levels of biofilm formation ability without summarizing the overall trends or insights, which could be streamlined for impact.

The antibiotic resistance section (lines 314-328) could be more clearly structured, particularly in differentiating between qualitative and quantitative screenings and in explaining the significance of the findings, such as why certain strains could grow on specific antibiotic plates and the implications of these results.

Again in the text initially states that 37 strains were isolated, but the resistance analysis mentions only 36 strains being tested against antibiotics, excluding one Plantibacter strain without explanation. This inconsistency in the number of strains considered in the resistance analysis needs clarification.

The summary states that rifampicin had inhibitory effects on all 36 strains, yet later sections classify some strains as multidrug-resistant bacteria (MDARB) without explaining how these strains are resistant to other antibiotics but not rifampicin. This contradiction requires clarification on the resistance profiles of the MDARB strains.

Discussion

The phrase "species-band bacteria" in line 411 is likely a typo and might confuse readers. It should be corrected to "seed-borne bacteria" to maintain consistency with the rest of the text and accurately reflect the subject of the study.

The hypothesis proposed in lines 422-425 regarding environmental pressure, bacterial abundance, and dominance is intriguing but is presented in a somewhat convoluted manner. Simplifying the explanation and clearly delineating the hypothesis would improve comprehension.

The statement in lines 448-450 suggests that the method used for classifying bacterial motility in this study has no clear basis, which undermines the reliability of the results. Providing a rationale for the chosen method or acknowledging its limitations and potential impacts on the findings would strengthen this section.

The discussion about antibiotics flowing into the environment through livestock excrement and affecting the microbiome (lines 456-458) is somewhat repetitive and could be condensed. This section reiterates well-known information without adding new insights specific to the study's findings.

The statement in lines 473-475 generalizes the ability of certain bacterial genera to colonize humans, animals, and environmental niches without providing specific evidence from the study to support these claims. Clarifying these points with study-specific data or acknowledging the need for further research would enhance the discussion's accuracy.

Comments on the Quality of English Language

Minor editing of English language required

Round 2

Reviewer 2 Report

Comments and Suggestions for Authors

NA

Comments on the Quality of English Language

Minor editing of English language required